# Effect of Dietary Casein Phosphopeptide Addition on the Egg Production Performance, Egg Quality, and Eggshell Ultrastructure of Late Laying Hens

**DOI:** 10.3390/foods12081712

**Published:** 2023-04-20

**Authors:** Wenpei Liu, Jun Lin, Chenyue Zhang, Zhi Yang, Haoshu Shan, Jiasen Jiang, Xiaoli Wan, Zhiyue Wang

**Affiliations:** 1College of Animal Science and Technology, Yangzhou University, Yangzhou 225009, China; 2Zhenjiang Animal Disease Prevention and Control Center, Zhenjiang 212000, China; 3Jurong Haoyuan Ecological Agriculture Technology Co., Ltd., Jurong 212400, China

**Keywords:** casein phosphopeptide, egg quality, eggshell ultrastructure, laying hen performance

## Abstract

(1) Objective: This study aimed to assess the effects of dietary casein phosphopeptide (CPP) supplementation on the egg production performance of late laying hens and the resulting egg quality and eggshell ultrastructure. (2) Methods: A total of 800 laying hens aged 58 weeks were randomly assigned into 5 groups with 8 replicates of 20 hens each. The hens were fed a basal diet supplemented with 0 (control, T1), 0.5 (T2), 1.0 (T3), 1.5 (T4), and 2.0 (T5) g/kg CPP for 9 weeks. (3) Results: Dietary CPP supplementation was found to be beneficial for improving eggshell quality. The spoiled egg rate of the experimental groups was lower than that of the control group (linear and quadratic effect, *p* < 0.05). The yolk color in the T2, T3, and T4 groups was higher than that in the T1 group (quadratic effect, *p* < 0.05). The shell thickness in the T4 group was higher than that in the T1 and T2 groups (linear effect, *p* < 0.05). The shell color in the experimental groups was higher than that in the control group (linear and quadratic effect, *p* < 0.05). The effective thickness in the T3–T5 groups (linear and quadratic, *p* < 0.05) and the number of papillary nodes in the T2 and T3 groups were higher than those in the T1 group (quadratic, *p* < 0.05). The calcium content in the T2 and T3 groups was higher than that in the T1 group (quadratic effect, *p* < 0.05). The iron content in the T2 and T3 groups was higher than that in the T1 group (*p* < 0.05). (4) Conclusion: In summary, 0.5–1.0 g/kg CPP supplementation reduced the spoiled egg rate, enhanced the yolk and eggshell colors, increased the thickness of the effective layer, and the calcium and iron contents in the eggshell.

## 1. Introduction

Eggshell quality has always been a key concern in egg production as eggs are a daily indispensable food for people. This is especially true in the late laying period when eggshell quality is significantly lower than in the early and peak laying periods, because eggshell quality is related to the age of the hens [1]. The excellent eggshell quality not only improves the hatching rate of breeding eggs, but also facilitates preservation and transportation, increasing economic benefits. At present, due to the high exploitation of egg production performance, the problem of declining eggshell quality has reached a bottleneck from a genetic point of view, so improvement from a nutritional point of view is possible. In recent years especially, experts have proposed improving egg-laying durability by extending the traditional 72 weeks of age to 80 or even 100 weeks of age to achieve the goal of “500 eggs at 100 weeks of age”. Thus, improving eggshell quality in later egg laying stages has become a popular research topic worldwide and a current scientific problem [2].

Currently, casein phosphopeptide (CPP) is a widely used nutritional enhancer for calcium absorption and is a low molecular weight bioactive peptide extracted from food-grade casein with a bioactive center of -SerP-SerP-SerP-Glu-Glu- and refined by protease hydrolysis. CPP can facilitate the combination of chlorine and calcium in a solid state, acting as a calcium carrier and increasing calcium uptake in teeth and bone. It has been proven to be a potential enhancer of calcium effectiveness in food, and is, therefore, commonly used in human food and health products [3]. Studies have shown that phosphoserine clusters in the environment of weak base can not only form soluble complexes with calcium, inhibit the formation of insoluble calcium precipitates, and promote the absorption of calcium in the small intestine, but also promote the absorption and utilization of trace elements such as iron and zinc in the same way [4]. As such, CPP is also known as a “mineral carrier”. However, the application of CPP in animal production is limited. According to a previous study, the addition of CPP in poultry diet is beneficial to performance, growth and development, and egg quality [5,6,7]. In addition, CPP in the low-calcium diet promotes calcification of epiphyseal cartilage in growing broilers [8]. This research focuses on egg production with different levels of CPP supplementation and, discusses the effects of different CPP supplementation levels on the egg production performance of late laying hens and the resulting egg quality and eggshell ultrastructure, thereby determining the suitable CPP supplementation level so that nutrition regulation technology can prolong the egg production period and improve eggshell quality.

## 2. Materials and Methods

### 2.1. Animals and Management

All animal experiments were conducted in accordance with the Institutional Animal Care and Use Committee of Yang Zhou University, Yang Zhou, China (No. SYXK(Su)2021-0020).

A total of 800 healthy Hy-Line Brown layers with similar weights at 58 weeks of age were obtained from Jurong Haoyuan Ecological Agriculture Technology Company Co., Ltd. (Jurong, China). In a well-ventilated closed coop, the chickens were raised in a stepped cage with 16 h/d artificial light, the temperature in the coop was 22~25 °C, and the relative humidity was 65~75%. The birds were randomly divided into 5 groups with 8 replicates of 20 birds each.

The dietary treatments included a basal diet (T1) and a basal diet supplemented with 0.5 (T2), 1.0 (T3), 1.5 (T4), and 2.0 (T5) g/kg CPP (provided by Linxia Hua An Biological Products Co., Ltd., Linxia, China). The basal diet was formulated in accordance with the new version of the Feeding Management Manual 2021 for Hy-Line Brown commercial laying hens and the NRC (1994) to satisfy the nutrient requirements of laying hens (Table 1). The total experimental period was 10 weeks, with a pretest period of 1 week and a formal period of 9 weeks. The hens were allowed free access to water throughout the experiment. The average daily feed consumption was adjusted to approximately 115 g/animal per day after entering the formal trial period. 

### 2.2. Sample Collection

The egg quality of 5 eggs collected randomly from each replicate at the end of the 9th week of the formal period was measured. At the end of the trial period, 1 hen was randomly selected from each replicate for slaughter, and after slaughter, dissection was performed to separate the follicles and count their numbers according to grade for subsequent data analysis, referring to the study of Gilbert et al. for follicle grade classification [9].

### 2.3. Performance and Egg Quality

To calculate egg production, egg weight was recorded daily for each replicate. The egg number, egg mass, and feed consumption were recorded weekly for each replicate. The laying rate, average egg weight (g), spoiled egg rate, and feed–egg ratio (total feed consumption divided by egg mass) were calculated.

The shell percentage was defined as the shell weight divided by the egg weight. The yolk percentage was defined as the yolk weight divided by the egg weight. Vernier calipers were used to determine the egg index (0–200 mm, Meinite Industrial Co., Ltd., Shanghai, China). The egg shape index was defined as the egg length divided by the egg width. Shell thickness was measured using a spiral micrometer to measure the thickness of the blunt end, middle and sharp end of the eggshell, and the result was expressed as an average (25–50 mm, Guilin Measuring Tool Cutting Tool Co., Ltd., Guilin, China). The shell strength was measured by a shell strength instrument (AC220, Orka Company, Shanghai, China). The eggs were placed vertically and bluntly upward when the measurement was determined, and the unit was N. The shell color was measured using a CR-400 color differential instrument (CR-400, Konica Minolta, Osaka, Japan), and the dull end, middle end, and sharp end of the eggshell were measured. The Haugh unit, the egg yolk color, and the egg weight were measured by egg quality measurement (EMT-7300, Robotmation Co., Ltd., Tokyo, Japan) with the following equation: Haugh unit = 100·log(H-1.7·W0.37 + 7.57).

### 2.4. Determination of Eggshell Ultrastructure

One egg from each group was selected from the eggs measured at week 9. The shell membrane was removed and dried at room temperature. The shells were trimmed into 0.5 cm × 0.5 cm squares and pasted on the observation table, gold-sprayed, and placed in a Gemini SEM 300 Zeiss field emission scanning electron microscope system for imaging (Gemini SEM 300, Carl Zeiss company, German). The thickness of each layer was measured with Image-Pro Plus 6.0. The papillary layer and the effective layer were measured three times each under the mirror image 150× magnification; the papillary width and the number of papillary nodules were measured under the mirror image at 200× magnification, and the number of papillary nodules per unit area was calculated.

### 2.5. Determination of Calcium, Phosphorus, and Iron in Eggshell

The calcium content was determined by the ethylenediaminetetraacetic acid (EDTA) method, the phosphorus content was determined by molybdenum yellow spectrophotometry, and the iron content was determined by the colorimetric method following the instructions of a commercial kit (Nanjing Jiancheng Bioengineering Institute, Nanjing, China).

## 3. Results

### 3.1. Performance

The performance results are shown in Table 2. The spoiled egg rate of the experimental groups was lower than that of the control group (linear and quadratic effect, *p* < 0.05).

Dietary CPP supplementation had no effects on the laying rate, egg weight, feed intake, or feed–egg ratio of laying hens.

### 3.2. Egg Quality

As shown in Table 3, the yolk color in the T2, T3, and T4 groups was higher than that in the T1 group (quadratic effect, *p* < 0.05), and there was no significant effect on the percentage of yolk (*p* > 0.05).

The shell thickness in the T4 group was higher than that in the T1 and T2 groups (linear effect, *p* < 0.05). The shell color of the experimental groups was higher than that of the control group (linear and quadratic effect, *p* < 0.05), and there was no significant effect on the shell percentage (*p* > 0.05).

As shown in Table 4, there were no significant differences in the Haugh unit egg shape index and protein height among all groups (*p* > 0.05).

### 3.3. Eggshell Ultrastructure

As shown in Table 5, the effect of increasing dietary CPP levels on the palisade layer thickness and vertical crystal layer thickness was linear (*p* < 0.05). The effective thickness of the 1.0, 1.5, and 2.0 g/kg CPP groups was higher than that of the control group (linear and quadratic effect, *p* < 0.05). The number of papillary nodes in the T2 and T3 groups was higher than that in the T1 group (quadratic effect, *p* < 0.05). The addition of CPP to the diet had no significant effect on the papilla layer thickness, vertical layer thickness, mammillary layer thickness, or width of the mastoid gap among the different groups (*p* > 0.05).

### 3.4. Number of Follicles

As shown in Table 6, the addition of CPP to the diet had no significant effect on the number of follicles in the later stages of egg production (*p* > 0.05).

### 3.5. Calcium, Phosphorus, and Iron Content of the Eggshell

As shown in Table 7, the calcium content in the T2 and T3 groups was higher than that in the T1 group (quadratic effect, *p* < 0.05). Dietary CPP supplementation had no significant effect on the phosphorus content (*p* > 0.05). The iron content in the T2 and T3 groups was higher than that in the T1 group (*p* < 0.05).

### 3.6. Statistical Analysis

The data were confirmed to follow a normal distribution by P-P graph test and tested by homogeneity of variance for one-way ANOVA. The CPP level was the independent variable, the observed data including egg performance, egg quality, eggshell ultrastructure, follicle diameter, and contents of calcium, phosphorus, and iron in the eggshell were the dependent variables. Duncan’s multiple range test and Dunnett were used for multiple comparisons. Polynomial orthogonal contrast was used to determine the linear and quadratic relationships between the dietary CPP level and the egg laying performance, egg quality, and eggshell ultrastructure. *p* < 0.05 was used to determine significant differences. *p* > 0.05 indicates no significant difference. Statistical analyses were carried out using SPSS version 22.0.

## 4. Discussion

### 4.1. Laying Performance

CPP is an active peptide that promotes calcium absorption and improves immune and reproductive performance. Studies have shown that the eggshell becomes thinner, the breakage rate increases, and the number of eggs laid increases after an egg-laying hen reaches 400 days of age [10]. CPP is a new product developed to improve calcium absorption and utilization. While vitamin D can promote the active transport absorption of saturated calcium in the upper small intestine, CPP promotes unsaturated passive diffusion absorption in the lower small intestine, and the passive diffusion absorption of calcium is much greater than the active transport absorption. Moreover, CPP is not affected by age and calcium intake. However, increasing calcium intake alone leads to urate deposition and interferes with the digestion, absorption, and utilization of other nutrients. Song et al. [11] showed that dietary supplementation with 0.5–1.5 g/kg calcium propionate can improve the laying performance of hens by improving calcium availability. According to previous studies, dietary CPP addition can greatly improve the growth performance and eggshell quality of poultry. In our study, it was found that there was no significant change in the egg production rate after dietary CPP supplementation. This may be due to differences in breed, age of chickens, and feeding management. Therefore, there was no significant difference in follicle diameter in this study. However, the shell breakage rate was reduced, and the improvement in eggshell quality may indicate the increased availability of minerals such as calcium required during eggshell formation.

### 4.2. Egg Quality

Yolk color, which is an important indicator when purchasing eggs, is influenced by yolk pigmentation, and eggs with a deep yolk color are preferred by consumers. A study showed that by regulating gastrointestinal pH and improving the intestinal digestive environment, the absorption of minerals such as calcium and phosphorus can be facilitated, which in turn promote the absorption, transport, and deposition of carotenoids, thereby deepening the yolk color [12]. In addition, InKee et al. [13] showed that the inclusion of different dietary iron sources improved yolk color. In the present study, the yolk color was significantly deepened, but no article has been reported on the change in yolk color due to dietary CPP addition, and the specific mechanism needs to be further investigated. On the one hand, CPP may promote the absorption of the trace element iron [14], thus enhancing the yolk color. On the other hand, CPP may promote the absorption of vitamins and other substances, and increase the deposition of carotenoids [12], thus enhancing the yolk color. Whether the change in yolk color has an effect on yolk nutrient content should be a future research direction, especially in the context of functional egg developments.

Elaroussi et al. [15] suggested that egg size and weight increase with chicken age, but shell weight generally does not increase proportionally with the aging of chickens, which leads to eggshell thinning and increasing shell breakage. In the Korean poultry feeding standard (2012), the calcium requirement for brown-shelled egg-laying hens is 4.1% at 110 g/d. A study [16] showed that a calcium level of 4.1% is not sufficient to maximize the shell quality and reduce the shell breakage rates for late laying hens. Therefore, it is necessary to increase the dietary calcium content or use dietary additives to promote intestinal calcium absorption. In vitro cellular experiments and in vivo animal experiments by Sun et al. [17] amply demonstrated the strong calcium absorption capacity was facilitated by CPP. Additionally, Parkinson et al. [18] showed that the chelation and solubility of dietary calcium and phosphorus in the intestine may be influenced by CPP, which stimulates the absorption of calcium and phosphorus and is able to act independently of vitamin D. Eggshell thickness increased, probably because CPP contains four negatively charged phosphorylated serine residues (Σ) which prevent the action of digestive enzymes and inhibit the formation of calcium phosphate precipitates. The binding properties of CPP and its resilience against protein hydrolysis promote calcium absorption. Once formed, this peptide–mineral complex becomes an effective carrier for calcium transport, maintaining a high free calcium concentration and promoting passive calcium absorption [19], thus increasing eggshell calcium deposition and alleviating eggshell thinning during the late egg-laying period. The Haugh unit is a conventional predictor of egg quality which is determined by protein height [20]. Studies have shown that the variety and feeding environment of laying hens can affect eggshell quality [21,22]. In this study, the addition of CPP promoted the calcium absorption of laying hens by improving the diet, independent of the feeding environment, so there was no significant difference in Haugh unit and protein height.

The eggshell color is the result of the secretion and pigmentation of the uterine glands and is an important indicator used by consumers to measure eggshell quality. The uniformity of the color not only reflects the purity of the egg-laying chicken breed, but also indicates the health of the flock and the egg quality. In the late egg-laying period, the eggshell color gradually becomes lighter as the age of the chickens increases, and eggs with a darker shell color are more popular among consumers, but there is no precise grading of the shade of brown-shell eggs [23]. It has been shown that the pigment in brown eggshell is formally protoporphyrin IX, and the concentration of iron, a raw material for heme synthesis, affects the production of protoporphyrin IX [24]. It was found that the addition of organic iron or ferrous sulfate in the diet of laying hens significantly improved eggshell color because iron is involved in the production of heme, which is closely related to eggshell pigmentation [13]. Therefore, in the present experiment, the enhancement of eggshell color might be because CPP has a role in promoting iron absorption, thus increasing the production and transport of protoporphyrin IX and heme and enhancing the deposition of pigments on the eggshell [24].

### 4.3. Eggshell Ultrastructure

The eggshell is an important barrier for material exchange between the internal environment of the eggshell and the external environment. The eggshell consists of an inner and outer shell layer, papilla layer, fenestra layer, vertical crystal layer, and a cuticle layer [25]. Among them, the fenestra and crystalline layers constitute the effective layer thickness [26]. A study showed that the effective layer of the shell, the ratio of the height of the fenestra layer to the papilla layer, and the width and density of the papilla nodes affect the strength of the shell [27]. The papillae layer is the site where the shell membrane protein fibers begin to calcify during shell formation, and whether the papillae are regularly arranged neatly between each other affects the formation of other layers and plays an important role in eggshell quality [28]. Low quality eggshells tend to have higher porosity or irregular arrangement. The mammillary layer is calcified from the nucleation sites on the outside of the outer membrane and finally forms basal parts of calcified columns and knobs [29]. In a study by Qiu et al. [26], the effective thickness and the density of the papillae nodes were found to play a decisive role in the strength of the eggshell. Moreover, the column of calcite crystals in the palisade layer and the thickness of this layer were found to be the factors affecting the fracture strength of the eggshell [30]. Eggshells are composed of 3 to 3.5% organic matter and 95% minerals, and egg quality is best when the proportion of organic and inorganic matter involved in eggshell formation is optimized [31]. In this study, the shell strength tended to increase with increasing palisade layer thickness. The reason for this may be that the addition of CPP promotes the formation of protein and other organic compounds in the eggshell, increasing eggshell toughness. However, whether CPP can promote the formation of proteoglycans and proteins in inorganic materials has not been reported and further research is needed. In addition, poorer eggshells with less strength tended to have larger papillary nodules, that is, fewer papillary nodules per unit area [32]. In the present study, the effective thickness and the number of papillary nodules increased with increasing CPP content, which further supported that the addition of CPP improved the eggshell ultrastructure.

Therefore, the increased shell thickness may be related to the variability in eggshell ultrastructure in the present study. At the same time, the increased effective layer height and the numbers of papilla nodules played an important role in improving eggshell strength, thus helping to alleviate the adverse effects of aging on the quality of eggs from laying hens.

### 4.4. Eggshell Calcium, Phosphorus, and Iron Content

Calcium is the most important nutritional factor affecting eggshell quality. A reduction in calcium deposition in eggshells leads to a decrease in eggshell quality and an increase in the eggshell damage rate. Wang et al. [33] found that intestinal digestion and absorption function decreased during the late laying period compared to the peak period. Therefore, it is necessary to improve the calcium absorption capacity of laying hens in the late stage of laying, strengthen the absorption and utilization of calcium through nutritional regulation, and alleviate abnormal eggshell calcium deposition, which is conducive to solving the problem of eggshell quality decline caused by increasing hen age [34]. CPP can chelate divalent calcium, iron, zinc, and other metal ions and protect them from precipitation by anions such as phosphoric acid, oxalic acid, and phytic acid in the diet to effectively promote their absorption and utilization [35]. With the increase of the CPP level, the Ca content increased. Although P content did not reach a significant difference, there was a trend of quadratic curve change, which proved that phosphorus indirectly affects eggshell quality through calcium and other trace elements. The change in eggshell color can indirectly reflect the health status of laying hens and the quality of eggs and is also the most intuitive indicator for consumers to evaluate the quality of eggs and whether to buy them. In this study, the deepening of eggshell color may be due to the increase of protoporphyrin IX, which increases the iron content in eggshell. Therefore, the increase in the iron content in the eggshell confirms the result that the addition of CPP in this experiment was beneficial for deepening the eggshell color. When CPP is combined with iron, the liver, spleen, and other organs involved in iron metabolism have a better absorption capacity for iron than inorganic iron. Ani Kibangou et al. [36] confirmed the bioactive role of CPP in iron bioavailability through CPP–Fe complex radiolabeling experiments. This suggests that the CPP–Fe complex can be a good candidate for iron transport. Therefore, in this study, the increased calcium and iron content in the eggshell may have been contributed to the promotion of calcium and iron absorption by CPP.

## 5. Conclusions

The results of the present study revealed that dietary CPP supplementation reduced eggshell breakage in laying hens, improved the eggshell and yolk color, increased the effective thickness and the number of papilla nodules in the eggshell, and increased the calcium and iron contents in the eggshell. In conclusion, dietary 0.5–1.0% CPP supplementation is appropriate for application in the late laying period.

## Figures and Tables

**Table 1 foods-12-01712-t001:** Basal diet composition and nutritional level (as-fed basis).

Ingredient	Content (%)
Corn	61.00
Soybean meal	25.00
Limestone	10.42
Wheat bran	2.00
Calcium hydrogen phosphate	0.78
Salt	0.35
DL-Methionine	0.08
Premix(Phytase) ^1^	0.37
Total	100.00
Nutritional levels ^2^	
Metabolizable Energy (MJ/kg)	10.83
Crude Protein (%)	16.30
Cacium (%)	4.02
Effective phosphorus (%)	0.34
Methionine (%)	0.34
Lysine (%)	0.83
Threonine (%)	0.61
Tryptophan (%)	0.18

^1^ The premix provided the following per kg of the diet: vitamin A, 10,000 IU; vitamin D3, 3000 IU; vitamin E (DL-αcophenol), 15 IU; vitamin K3, 2.0 mg; vitamin B1, 1 mg; vitamin B2, 4.5 mg; vitamin B6, 3 mg; vitamin B12, 0.015 mg; nicotinamide, 25 mg; pantothenic acid; 4.5 mg; folic acid,0.75 mg; choline chloride, 300 mg; Fe (ferrous sulfate), 70 mg; Mn (manganese sulfate), 75 mg; Zn (zinc sulfate), 80 mg; I (calcium iodate), 0.6 mg; Cu (copper sulfate), 8 mg; Se (sodium selenite), 0.3 mg. ^2^ Nutritional ingredients are calculated values.

**Table 2 foods-12-01712-t002:** Effects of casein phosphopeptide (CPP) on the performance of laying hens.

Item ^1^	T1	T2	T3	T4	T5	SEM	*p*-Value	
							Linear	Quadratic
Laying rate (%)	82.76	83.71	84.59	78.91	80.34	0.872	0.117	0.435
Egg weight (g)	65.10	65.36	64.63	64.97	64.88	0.146	0.431	0.766
Feed intake (g/hen/d)	113.35	113.31	113.02	111.68	112.01	0.337	0.078	0.915
Feed–egg ratio	2.15	2.12	2.11	2.22	2.21	0.028	0.275	0.486
Spoiled egg rate (%)	5.09 ^a^	3.42 ^bc^ **	2.75 ^c^ ***	3.67 ^bc^ *	3.85 ^b^ *	0.186	0.046	<0.001

^a–c^ Values within a row with different superscripts differ significantly at *p* < 0.05., while with the same or no letter superscripts mean no significant difference (*p* > 0.05). *p* < 0.05 was marked with *, *p* < 0.01 was marked with **, *p* < 0.001 was marked with ***. ^1^ Laying hens in T1, T2, T3, T4, and T5 groups were fed a basal diet (without additional casein phosphopeptide) supplemented with 0 (control), 0.5, 1.0, 1.5, and 2.0 g/kg casein phosphopeptide, respectively. The result is the mean value and SEM is the standard error of the mean value, n = 8.

**Table 3 foods-12-01712-t003:** Effects of casein phosphopeptide (CPP) on the eggshell quality of laying hens.

Item ^1^	T1	T2	T3	T4	T5	SEM	*p*-Value	
							Linear	Quadratic
Shell strength(N)	4.50	4.65	4.68	4.70	4.74	0.042	0.083	0.522
Shell thickness(um)	324.10 ^b^	325.10 ^b^	332.70 ^ab^	341.60 ^a^ *	337.50 ^ab^	2.260	0.006	0.618
Shell color	16.25 ^c^	17.72 ^b^ ***	19.29 ^a^ ***	19.40 ^a^ ***	18.51 ^ab^ ***	0.235	<0.001	<0.001
Percentage of shell (%)	10.81	10.78	10.66	10.56	10.59	0.060	0.134	0.789
Percentage of yolk (%)	26.66	27.20	27.08	27.20	27.69	0.199	0.162	0.930
Yolk color	7.13 ^bc^	7.59 ^a^ *	7.35 ^ab^	7.33 ^ab^	7.01 ^c^	0.055	0.133	0.002

^a–c^ Values within a row with different superscripts differ significantly at *p* < 0.05, while with the same or no letter superscripts mean no significant difference (*p* > 0.05). *p* < 0.05 was marked with *, *p* < 0.001 was marked with ***. ^1^ Laying hens in T1, T2, T3, T4, and T5 groups were fed a basal diet (without additional casein phosphopeptide) supplemented with 0 (control), 0.5, 1.0, 1.5, and 2.0 g/kg casein phosphopeptide, respectively. The result is the mean value and SEM is the standard error of the mean value, n = 40.

**Table 4 foods-12-01712-t004:** Effects of casein phosphopeptide (CPP) on Haugh unit, protein height, and egg shape index of laying hens.

Item ^1^	T1	T2	T3	T4	T5	SEM	*p*-Value	
							Linear	Quadratic
Haugh unit	82.27	82.46	85.09	84.38	82.93	0.566	0.426	0.172
Egg-shaped index	1.29	1.30	1.30	1.30	1.30	0.002	0.189	0.371
Protein height	7.01	7.07	7.45	7.44	7.09	0.094	0.446	0.139

With the same or no letter superscripts mean no significant difference (*p* > 0.05). ^1^ Laying hens in T1, T2, T3, T4, and T5 groups were fed basal diet (without additional casein phosphopeptide) supplemented with 0 (control), 0.5, 1.0, 1.5, and 2.0 g/kg casein phosphopeptide, respectively. The result is the mean value and SEM is the standard error of the mean value, n = 40.

**Table 5 foods-12-01712-t005:** Effects of casein phosphopeptide (CPP) on ultrastructure of eggshell of laying hens.

Item ^1^	T1	T2	T3	T4	T5	SEM	*p*-Value	
							Linear	Quadratic
Mammillary layer thickness/um	214.02	184.77	199.39	196.75	198.92	3.597	0.461	0.124
Palisade layer thickness/um	185.92	191.60	203.17	212.56	205.63	3.442	0.013	0.320
Vertical crystal layer/um	60.79	74.01	79.52	77.07	76.73	2.413	0.038	0.075
Effective thickness/um	246.72 ^c^	265.61 ^bc^	282.69 ^ab^ **	289.63 ^a^ **	282.37 ^ab^ **	3.937	0.001	0.027
Width of mastoid gap/um	99.46	93.60	88.41	97.40	96.82	4.133	0.963	0.514
Nipple nodules number/one	8.20 ^b^	9.80 ^a^ *	10.40 ^a^ **	9.40 ^ab^	9.40 ^ab^	0.224	0.143	0.006

^a–c^ Values within a row with different superscripts differ significantly at *p* < 0.05, while with the same or no letter superscripts mean no significant difference (*p* > 0.05). *p* < 0.05 was marked with *, *p* < 0.01 was marked with **. ^1^ Laying hens in T1, T2, T3, T4, and T5 groups were fed basal diet (without additional casein phosphopeptide) supplemented with 0 (control), 0.5, 1.0, 1.5, and 2.0 g/kg casein phosphopeptide, respectively. The result is the mean value and SEM is the standard error of the mean value, n = 8.

**Table 6 foods-12-01712-t006:** Effects of casein phosphopeptide (CPP) on the follicle number of laying hens.

Follicle Diameter (mm) ^1^	T1	T2	T3	T4	T5	SEM	*p*-Value	
							Linear	Quadratic
1 ≤ d < 4	48.75	39.00	43.25	49.37	45.62	2.031	0.777	0.423
4 ≤ d < 5	12.75	10.00	10.00	11.00	12.75	0.705	0.844	0.103
5 ≤ d < 8	10.37	11.87	9.62	10.50	11.75	0.626	0.765	0.630
8 ≤ d < 40	5.75	5.25	5.00	5.25	5.37	0.144	0.476	0.164

With the same or no letter superscripts mean no significant difference (*p* > 0.05). ^1^ Laying hens in T1, T2, T3, T4, and T5 groups were fed basal diet (without additional casein phosphopeptide) supplemented with 0 (control), 0.5, 1.0, 1.5, and 2.0 g/kg casein phosphopeptide, respectively. The result is the mean value and SEM is the standard error of the mean value, n = 8.

**Table 7 foods-12-01712-t007:** Effect of casein phosphopeptide (CPP) on calcium, phosphorus, and iron content in eggshell.

Item ^1^	T1	T2	T3	T4	T5	SEM	*p*-Value	
							Linear	Quadratic
Ca/%	32.85 ^c^	35.72 ^a^ *	35.47 ^ab^	33.27 ^bc^	33.62 ^abc^	0.376	0.708	0.021
P/g/kg	1.20	1.21	1.34	1.27	1.19	0.025	0.941	0.073
Fe/mg/kg	3.79 ^bc^	5.05 ^ab^	5.60 ^a^ *	3.31 ^c^	5.42 ^a^ *	0.266	0.272	0.362

^a–c^ Values within a row with different superscripts differ significantly at *p* < 0.05, while with the same or no letter superscripts mean no significant difference (*p* > 0.05). *p* < 0.05 was marked with *. ^1^ Laying hens in T1, T2, T3, T4, and T5 groups were fed basal diet (without additional casein phosphopeptide) supplemented with 0 (control), 0.5, 1.0, 1.5, and 2.0 g/kg casein phosphopeptide, respectively. The result is the mean value and SEM is the standard error of the mean value, n = 8.

## Data Availability

Data are available on reasonable request from the corresponding author.

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
