# Peer review of "Effect of Dietary Casein Phosphopeptide Addition on the Egg Production Performance, Egg Quality, and Eggshell Ultrastructure of Late Laying Hens"

_foods, 2023, doi:10.3390/foods12081712_

Round 1
Reviewer 1 Report
Comments and Suggestions for Authors
Manuscript: Effect of dietary casein phosphopeptide addition on the egg 2 production performance, egg quality and egg shell ultrastructure of late laying hens
2. Materials and Methods
2.6. Statistical analysis
The statistical analysis section is incomplete, as the authors do not clarify whether they validated the assumptions that allow the use of one-way ANOVA.
It is not clear which variable is being studied in the analysis of variance, as well as which factor is being compared. Authors must clarify.
What justifies the use of Duncan's test for multiple comparisons? On the other hand, since there is a control group, then multiple comparisons in that case should be performed with the Dunnett test.
Information about the software used is something that should come at the end of the statistical analysis section and not at the beginning.
3. Results
What does "SEM" mean in table 2? It should be in the table caption. Authors must indicate the value of the standard deviation for each of the situations.
The value of "n" is missing in the legend of the tables.
Line 144: When there are no statistically significant differences, the authors must indicate that p-value>0.05 (as shown later in the text).
Author Response
Dear editors,
On behalf of my co-authors, we appreciate editor and reviewers very much for their positive and constructive comments and suggestions on our manuscript entitled "Effect of dietary casein phosphopeptide addition on the egg production performance, egg quality and egg shell ultrastructure of late laying hens". Those comments are all valuable and very helpful for revising and improving our paper, as well as the important guiding significance to our researches. We have studied comments carefully and have made correction. Revised portion are marked in yellow highlighting in the paper. During the inspection, a clerical error was found in the article, which has been corrected. We hope the revised manuscript is now suitable for publication on《Foods》:
Thank you very much for your time and consideration.
Yours Sincerely,
Wenpei Liu

Reviewer 2 Report
Comments and Suggestions for Authors
Major comments
In general, research methodology is well designed and described. However, there is more available literature on the specific topic (in both broilers and laying hens). Please add more references in Introduction and Discussion sections, especially in Introduction section, which is relatively short in order to explain the rationale.
In Discussion section, all parameters examined should be discussed, even those found not statistically important. Moreover, if there are differences between your study and previous studies, you should provide possible explanations for any inconsistencies.
L223-225: Add references for the previous studies and present their findings in more detail.
Minor comments
L36-37: Rephrase. The sentence between the two commas (“it is also…people”) refers to the egg, not the eggshell.
L37: Remove the comma (,).
L56-57: Add a reference. Which is the previous study?
L57-59: Rephrase.
L61: Remove the word “different”, as it is written twice.
In Tables, it is recommended to explain the abbreviation CPP.
L259: Correct the phrase “its function as a calcium”.
L320-323: The content of these lines has already been explained previously.
Author Response

(The authors gave the same response as above.)

Round 2
Reviewer 2 Report
Comments and Suggestions for Authors
The manuscript has been improved, although some points would have deserved some more attention.